# The Struggle to Entertain Yourself: Consequences of the Internal Stimulation Factor of Boredom Proneness during Pandemic Lockdown

**DOI:** 10.3390/bs12090303

**Published:** 2022-08-25

**Authors:** Van Dang, Heather C. Lench

**Affiliations:** Department of Psychological and Brain Sciences, Texas A&M University, College Station, TX 77845, USA

**Keywords:** boredom, emotion, pandemic, boredom proneness

## Abstract

Boredom is a ubiquitous human experience that most people try to avoid feeling. People who are prone to boredom experience negative consequences. This study examined the impact of individual differences in the ability to entertain the self (the internal stimulation factor) on boredom experiences during the COVID-19 lockdown in the United States. The internal and external stimulation factors predicted greater boredom frequency, boredom duration, and boredom intensity, each of which reflected a different aspect of emotional experience. The relationship among these factors was complex. A serial mediation analysis indicated the internal stimulation factor predicted the frequency of boredom, which in turn predicted the duration of boredom, which predicted boredom intensity. This pattern of relationships is potentially unique to boredom among emotional experiences. These findings provide insight into how boredom functions during a period in which daily activities and coping resources that would normally be available became severely limited.

## 1. Introduction

All people experience boredom to greater or lesser degrees [1]. Despite the ubiquity of boredom in human experience, people express a desire to avoid feeling bored, and the emotion is frequently associated with unpleasant physiological arousal [1,2,3]. Boredom is experienced as aversive, but appears to serve several important functions. Bench and Lench [4] argued that boredom as an emotion motivates people to change their state and situation through seeking new experiences, and boredom has been conceptualized as a self-regulatory emotion because of this function [5]. Boredom is theorized to arise when people are in a situation that does not evoke another emotional response, such as happiness or anger or sadness [4]. That lack of emotional response means that the situation is not attracting their attention or engagement, and these situations have been classified as being not satisfying, perceived as meaningless, or without sufficient challenge [6,7,8,9,10].

The initial lockdown phase of the COVID-19 pandemic in the United States presented an unusual situation in which to examine the experience of boredom. During this period, every state imposed the same social isolation requirements in an effort to protect public health and safety. Normal daily activities were severely restricted and most people were confined to their own homes with limited social or work activity. From an emotional perspective, this situation limited people’s options for what they could do and explore when they experienced boredom. As a result, people’s ability to entertain themselves, often through engagement with their own thoughts and interests, was likely to be particularly important for alleviating the experience of boredom. The present investigation focused on the relationships between external versus internal stimulation boredom proneness and characteristics of boredom experience, including the frequency with which people felt bored, the duration of their boredom, and the experienced intensity of boredom.

### 1.1. External and Internal Stimulation Boredom Proneness

People vary in their experiences of boredom, and this individual difference has been captured through the measurement of boredom proneness. It is important to note the distinction between state and trait boredom. State boredom is defined as an aversive emotion that encourages pursuit of alternative goals and experiences [4]. Like most emotions, it tends to be short-lived and a response to a specific situation or circumstance. Boredom proneness, also termed trait boredom, represents variation in the frequency of experiences of boredom, and is associated with detrimental outcomes, such as gambling [11], depression [12,13,14], anxiety [12,13], substance abuse [12], noisy decision making irrespective of risk level [15], and risk taking [16]. Boredom proneness is most commonly measured by the Boredom Proneness Scale (BPS) [17]. This scale has been conceptualized as measuring a unitary construct, although there is evidence that the scale could be measuring two separate factors [18,19].

The two factors of boredom proneness were proposed by Vodanovich and colleagues [20] based on factor analysis of the boredom proneness scale that revealed an “external stimulation factor” and an “internal stimulation factor” [18,19,20,21]. The external stimulation factor represents individuals’ ability to fulfill heightened needs for excitement, change, and challenge in their environment. The internal stimulation factor captures individuals’ ability to maintain interest and engagement [19]. In other words, people with higher scores on the external stimulation factor of boredom proneness possess high needs for external stimulation, but often fail to take actions to fulfill that need or perceive difficulty in doing so. Those with high scores on the internal stimulation factor fail to generate interest and keep themselves entertained in response to boredom or perceive difficulty in doing so.

There are ongoing disagreements about the utility of examining two factors within the BPS [22,23], but there does appear to be incremental predictive validity in examining two factors [24]. The internal and external stimulation factors of trait boredom proneness predict different self-regulatory approaches [25]. Participants with higher scores on the external stimulation factor of boredom proneness exhibited lower success in avoiding losses and approaching non-losses. In contrast, those with higher scores on the internal stimulation factor demonstrated lower success in engagement on goal-related tasks and generating alternative solutions.

Given the restrictions imposed by the COVID-19 lockdown in the United States, where daily activities such as work and social engagement were severely limited, the internal stimulation factor of boredom proneness is potentially relevant to people’s experience of boredom. In a more typical environment, outside of a lockdown, people who chronically struggle to maintain interest and engagement in their thoughts and activities have other, external options that can help keep them entertained and stave off boredom. That ability to engage other activities was limited during the pandemic shut down. The initial pandemic lockdown presented a conundrum for these individuals. Their inability to stay interested and engaged, coupled with significantly limited options for external engagement, could have worsened the negative impact of isolation. In other words, those high on the internal stimulation factor of boredom proneness may struggle to identify alternative solutions to alleviate feelings of boredom, thus exacerbating their boredom experience, such as increased frequency of boredom, longer boredom duration, and increased boredom intensity.

### 1.2. Facets of Boredom Experience

Emotional experiences are complex and evidence has shown they are composed of multiple facets of experience that people are capable of differentiating in their self-reports [26]. These facets have not been previously investigated in studies of boredom, but, to the extent that boredom reflects an emotional experience, they are likely to represent unique aspects of the experience. Three major facets of emotional experience are frequency, duration, and intensity. While these three facets can co-occur during and after an event or thought, they appear to have unique precedents and consequences. One way to conceptualize these facets is as similar to aspects of a musical note. There is the frequency with which a note occurs in a sequence, representing unique instances of that note. There is the duration of a note, in terms of how long the note is held. There is also the intensity of the note, representing how loud or softly it is perceived. Similarly, emotional experiences can vary in how frequently they occur, how long they last, and their intensity.

The frequency of an emotional experience represents how often that emotion occurs during a period of time. People’s thought content typically changes rapidly and, partially as a result of this variability, people’s emotional experiences also fluctuate even in response to a singular event. Experiences of happiness, for example, might occur after a success and then fade, but will re-occur when thinking about that success later or when later recounting the event socially [27]. Similarly, boredom is likely to fluctuate over time and people are likely to vary in the frequency of their experiences of boredom. Questions in the Boredom Proneness Scale appear to best capture the frequency of boredom experience. Consistent with this, findings demonstrating a relationship between boredom frequency and boredom proneness [28,29].

The duration of an emotional experience represents the perceived length of time that the emotion lasts, and can last from seconds to hours [30]. People typically cope with and adjust to emotions fairly rapidly after they occur, with people’s state returns to baseline, even for impactful events [31,32]. Interestingly, the frequency of thoughts about an event or experience, and the frequency of the emotions that co-occur with those thoughts, are potentially predictive of the duration of an emotion. For several types of commonly experienced emotions, such as joy and sadness, more frequent related experiences predicted a longer duration of the experience [30]. In other words, the more that people think and feel about an experience, the longer that experience lasts. The relationship between the frequency and duration of experience has not been explored for boredom, but this previous work suggests that when people experience frequent boredom that the experience of boredom will also be perceived as lasting longer and be maintained for greater length of time. Although duration of boredom is a particularly pertinent characteristic of boredom experience, it has rarely received attention in the boredom proneness literature. Several studies have investigated the impact of boredom proneness on perceived passage of time, and found that boredom prone individuals perceive time as passing by more slowly during boring tasks [33].

The intensity of an emotional experience represents the strength of the response. Past studies have demonstrated that the intensity of experienced emotion is strongly related with the perceived importance of the event that evoked emotion [34,35]. This relationship is potentially different from other states in the case of boredom, as boredom is theorized to be evoked by a lack of other emotional responses. Thus it is the perceived absence of response that is being evaluated for importance, and this experience is likely to be perceived as more important when it is frequent and of long duration. This could result in a cascade of boredom responses, in that boredom that is frequent and long-lasting is also perceived as important and thereby associated with the experience of intense boredom. Previous studies have linked boredom proneness with boredom intensity [7,29], including in experience-sampling studies [36,37] and experimental contexts [38]. What is not currently understood is how experienced boredom intensity relates to the frequency and duration of boredom experiences.

There is ample evidence to support that emotions are characterized by the frequency, duration, and intensity of experience. These characteristics and their relationship, as well as any associations with boredom proneness, have not been explored for boredom.

### 1.3. The Consequences of Boredom Proneness in the COVID-19 Pandemic Lockdown

In the midst of the COVID-19 lockdown period, businesses were mostly closed and people were confined to their homes. During this time, boredom became an increasingly prevalent emotional experience to most people [39,40,41,42,43]. Several studies demonstrated associations between boredom proneness and undesirable outcomes during the early stages of the COVID-19 pandemic, including reduced adherence to social distancing guidelines [39,40,42]. Moreover, the ability to engage in self-directed activity, such as creative endeavors, was associated with lower levels of depression, anxiety, and boredom proneness [44]. This finding suggests that the internal stimulation factor of boredom proneness, and the associated ability to entertain the self, might be particularly important for responses during this lockdown period.

It is worth noting that recognizing the lack of meaning or value of a task, and experiencing boredom, is not problematic in and of itself. It is the failure to identify and launch into alternative actions that makes boredom such an unpleasant experience [45]. Problems arise when the boredom prone continuously fail to identify goals or take actions that they deem as meaningful and satisfying. Unfortunately, the unusual landscape of the COVID-19 pandemic lockdown presented people with extremely limited options and resources to cope with boredom.

### 1.4. The Present Investigation

The current study explored the relationship between the two factors of boredom proneness and other characteristics of boredom experiences, including boredom frequency, boredom duration, and boredom intensity. This study took place during the initial phase of the pandemic lockdown in the United States wherein every state imposed the same social isolation policies. Therefore, the aim of this study is to examine whether the internal stimulation factor of boredom, which represents the ability to keep the self entertained and engaged, would be particularly important for the experience of boredom. Given the exploratory nature of the study, we did not declare any a priori hypotheses.

## 2. Materials and Methods

### 2.1. Participants

Participants were recruited via a social media platform to complete an online survey. Data collection occurred during the initial phases of the COVID-19 shut-downs across states, which disrupted people’s lives and in many cases their internet services. The final sample consisted of 66 participants, and 27 participants were removed due to missing data (more than 85% of the survey). Therefore, the final sample consisted of 39 participants (73.5% female, 20.6% male, 5.9% responding as “other”) with a mean age of 28.2 years (*SD* = 13.3). Participants identified as White (79.4%), Asian (8.8%), Hispanic (5.9%), multiracial (2.9%), and other (2.9%). Participants were informed that the study was about activity when options were limited, and that they would be entered into a raffle for a gift certificate once they completed the survey.

### 2.2. Procedure

Participants completed the online survey about their experiences. The survey took approximately 20 min to complete.

#### 2.2.1. Boredom Experience

Participants responded to questions that assessed the intensity, duration, and frequency of their feelings of boredom, based on questions used previously in the emotion literature to measure these features of experience [26]. Specifically, participants were prompted, “Please rate how intense you are feeling the following emotions right now—Boredom,” on a scale ranging from not at all (1) to extremely (7). Participants were then asked, “Overall, how much of the time today did you feel the following emotions?—Boredom,” to which they rated their duration of boredom using a scale ranging from not at all (0) to the entire day (100). Lastly, participants rated the frequency of their boredom, “How frequently today did you feel bored?” on a scale ranging from not at all (1) to almost constantly (9).

#### 2.2.2. Boredom Proneness

Individual differences in the frequency of boredom experience were measured using the Boredom Proneness Scale [17]. Participants rated, using a scale ranging from (1) strongly disagree to strongly agree (7), the degree to which the 28 items were typical for them (i.e., “I am often trapped in situations where I have to do meaningless things”). High scores on the scale indicate greater propensity to experience boredom. In addition to composite scores of boredom proneness, participants’ ratings were further divided into an external factor (a perceived lack of external stimulation) and an internal factor (a perceived lack of internal stimulation). Based on previous factor analyses of the BPS [18,19,20,21], items 5, 6, 9, 10, 12, 15, 17, 19, 20, 21, 25, 26, 27, and 28 (i.e., “Many things I have to do are repetitive and monotonous”) were included in the external factor analysis; whereas the internal factor included items 1, 7, 8, 13, 18, 22, 23, and 24 (i.e., “It is easy for me to concentrate on my activities”).

## 3. Results

As our sample size was smaller than ideal given the COVID-19 situation and our study was exploratory, we highly recommend readers focus on the magnitude/effect sizes of the relationships rather than focusing solely on traditional levels of significance. The small sample means that only large effect sizes are likely to reach traditional thresholds for statistical significance, whereas small to medium effect sizes are unlikely to reach that threshold. We conducted a sensitivity power analysis for a bivariate correlational model using G*Power [46], and based on our sample size (*N* = 39), power of 0.8, and alpha of 0.05, effect sizes above 0.48 are adequately powered with our sample.

### 3.1. Boredom Proneness and Boredom Experience

We examined the relationships among trait boredom proneness, the internal stimulation and external stimulation factors of boredom proneness, boredom frequency, boredom duration, and boredom intensity. Generally, correlations of 0.10 are considered small, correlations of 0.30 are considered medium, and correlations of 0.50 are considered large [47]. As shown in Table 1, trait boredom proneness was positively correlated with the internal stimulation and external stimulation factors of boredom, boredom duration, and boredom frequency. Interestingly, results revealed that the internal stimulation factor of boredom was significantly associated with boredom frequency, while the external stimulation factor of boredom proneness was significantly associated with boredom duration and boredom frequency. This pattern supports assertions that the internal and external factors of boredom proneness are best considered independently.

#### 3.1.1. Predictors of Boredom Experience

We conducted multiple linear regression analyses to assess whether the two factors of boredom proneness predicted each facet of boredom experience while controlling for other boredom experiences. These analyses build on the correlational findings by simultaneously accounting for the variance in outcomes associated with other boredom experiences.

Results for boredom duration are shown in Table 2, and demonstrated that boredom intensity and boredom frequency were both associated with longer boredom duration. The overall model fit was significant, *F*(4, 26) = 24.83, *p* < 0.001, *adjusted R*^2^ = 0.76. Importantly, while boredom duration was associated with the intensity and frequency of boredom, the pattern of results suggests that these experiences can be differentiated for boredom, as is the case for other emotions.

As shown in Table 3, the internal stimulation factor of boredom proneness predicted higher frequency of boredom, whereas the external stimulation factor of boredom was not associated with boredom frequency. The duration of boredom was associated with higher boredom frequency, whereas boredom intensity was not associated. The overall model fit was significant, *F*(4, 26) = 16.80, *p* < 0.001, *adjusted R*^2^ = 0.68. This pattern of results suggests that frequency is especially relevant for experiences of boredom duration and, at least within the context of the pandemic lockdowns, associated with difficulty entertaining the self.

Finally, results for boredom intensity are shown in Table 4. Notably, boredom duration predicted greater boredom intensity, whereas boredom frequency did not. The overall model fit was significant, *F*(4, 26) = 10.54, *p* < 0.001 *adjusted R*^2^ = 0.56. This pattern again suggests that boredom experiences are differentiated according to their frequency, duration, and intensity.

#### 3.1.2. Mediation Model

Previous analyses revealed that the internal stimulation factor of boredom proneness predicted boredom frequency (Table 3), boredom frequency predicted duration of boredom (Table 2), and duration of boredom predicted boredom intensity (Table 4). Given this pattern, we explored the possibility of serial mediation using the PROCESS 4.0 package by Hayes [48] using Model 6 and 5000 bootstrap samples. As shown in Figure 1, the results supported serial mediation. The indirect effect associated with the sequential path of the internal stimulation factor of boredom proneness predicting boredom frequency, which predicted the duration of boredom, which in turn predicted boredom intensity, was significant (*β* = 0.07, 95% CI [0.02, 0.14]). This pattern of results is intriguing and suggests a potentially unique pattern for the emotion of boredom, whereby the frequency, duration, and intensity of boredom are associated with one another.

## 4. Discussion

The COVID-19 pandemic lockdown led to a flourishing of boredom experiences, and this may have been exacerbated among people who were already prone to boredom. The unusual situation of a pandemic lockdown left people with limited options for social and work activity, two common ways that people cope with boredom and keep themselves occupied. We theorized that, in this type of situation, people’s ability to entertain themselves would be particularly important for coping with and reducing the aversive emotion of boredom.

### 4.1. External and Internal Stimulation Boredom Proneness

Boredom proneness was initially conceptualized as a single construct, with higher scores representing a greater susceptibility to the experience of boredom [17]. Subsequent work suggested that boredom proneness actually captured two separate constructs: an external stimulation factor that represented a need for change and excitement in the environment, and an internal stimulation factor that represented the ability to maintain interest and engagement [19]. The present investigation contributed to the current literature and lent support to the proposed two factors of boredom proneness, such that the internal stimulation factor of boredom proneness predicted boredom experiences during the initial phase of pandemic lockdown. This finding is consistent with arguments that boredom proneness represents two separate factors, and reveals predictive incremental validity of assessing the two factors separately. It is important to note that this was during an unusual situation where people’s options for external entertainment were artificially restricted for long periods of time. Future research should examine whether the two factors of boredom can be differentiated in terms of their effects during more typical situations as well as any boundary conditions that restrict the ability to differentiate the two factors.

While many studies have investigated the impact of boredom proneness on functioning, boredom proneness as a construct remains unclearly conceptualized. Boredom proneness has been theorized to reflect a personality trait that predispose people to experience boredom more frequently [17]. Nonetheless, boredom proneness may still be influenced by situational contexts, meaning that external circumstances coupled with individual differences in boredom proneness could impact people’s general experience of boredom. Given that the external and internal stimulation factor of boredom proneness had different relationships with boredom experience during this lockdown period, and the internal stimulation factor most strongly related to responses, it may indicate that there is indeed a meaningful difference between these two factors, especially under severe external constraints wherein people are left without many choices to cope with boredom.

### 4.2. Facets of Boredom Experience

Previous studies of emotions, such as happiness and anger, have revealed that people can and do differentiate the frequency, duration, and intensity of emotional experiences [26]. The present investigation revealed that people similarly differentiate these characteristics for the experience of boredom and that understanding each of these characteristics is likely necessary to develop a reasonable theory regarding boredom and its consequences.

Similar to past conceptualizations and findings, the present investigation revealed a link between the internal stimulation boredom proneness and the frequency of boredom [28,29,49]. The relationship between boredom proneness and the duration and intensity of boredom was more complex. A serial mediation analysis revealed that the internal stimulation factor of boredom proneness predicted boredom frequency, which in turn predicted the duration of boredom, which predicted the intensity of boredom. This path is potentially unique to boredom among emotional experiences. Boredom has been theorized to arise from a lack of other emotional responses, such as happiness, anger, and sadness, and that boredom then prompts people to explore and seek new experiences that do elicit emotions. Unlike other emotions, the focus of boredom is not on a situation in the environment (such as the success that leads to happiness or the loss that leads to sadness), but instead the internal environment of a lack of response. The intensity of emotion is closely linked to the perceived importance of the event [34,35]. As boredom is focused on the internal environment, the frequency and duration of boredom could be relevant signals about the importance of the internal environment. In other words, it is as if people are observing they are frequently bored for long periods and then that feels very important and aversive, and then the experience of boredom becomes more intense. The cumulative effect of these interactions may gradually result in increased boredom intensity, and potentially are the reason that people high in boredom proneness seek out activities that are stimulating but detrimental, such as gambling [11] and substance use [12].

Our findings provide insight into the need to assess the duration of boredom. People differentiated the duration of boredom from the frequency or intensity of boredom. Boredom is sometimes characterized by a sense of being “inescapable” and this feature of the length of boredom could reflect people’s ability to cope with and change their state of boredom. Feeling “stuck” in boredom might be particularly aversive, and potentially this characteristic could be particularly important for entering states such as depression and anxiety that can occur with boredom proneness.

## 5. Limitations and Future Directions

The intensity of an emotional experience is partly dependent on an individual’s evaluations of the importance of events in relation to their life goals. Research on subjective well-being posits that people tend to adapt to their external circumstances, i.e., [31,50,51], which enables them to return their emotions to baseline levels [31,32,52]. The COVID-19 pandemic lockdown was a unique landscape that enabled state boredom to thrive. People did not have access to work and social situations that are common ways to engage and keep the self entertained. However, people do quickly adapt to and cope with the constraints of their external environment, and it is possible that the intensity with which they experienced boredom should decrease over time. With respect to boredom prone individuals who perceive difficulty in maintaining engagement and interest, it may be the case that they experienced boredom more frequently as a result of ineffective coping, thus increasing the time of their boredom episode. The links between boredom proneness and coping behavior is an important direction for future research.

It is imperative to understand how boredom functions because boredom is an aversive emotional experience that most people want to avoid. Once people become aware of this unpleasant emotional experience, they will likely employ coping strategies aimed to alleviate boredom. Coping strategies in response to feeling bored can range from the innocuous, such as daydreaming and motor restlessness [53] to the problematic, including emotional eating [54,55,56], gambling [11], and substance abuse [12]. However, boredom is not all bad as it prompts people to take actions and find alternative solutions to their current undesirable circumstances. For instance, when a college student constantly feels bored and unmotivated when completing their coursework, boredom may signal to this individual that their current major is unfulfilling and unsatisfying. Thus, boredom in this context may encourage this person to perhaps evaluate their life meaning, look into other career options, and take actions accordingly. As a result, understanding the characteristics that represent boredom proneness will further our understanding as to how boredom functions across individuals. More specifically, future studies can examine ways in which characteristics of boredom proneness influence people’s choice of coping strategies.

There are some notable limitations to our study. Our sample size was relatively small, thus reducing the overall power of the study. Further, our study primarily relied on self-reports and was strictly correlational in nature. Thus, an experimental research study is needed to establish causal links between boredom proneness and other boredom experiences. One limitation of our study is that only the Boredom Proneness Scale was used, instead of the recent Short Boredom Proneness Scale [23] and the Boredom Proneness Scale—Short Form [20,57]. It would be useful to examine whether similar patterns emerge when using a different measure of boredom proneness. Given that our data was specifically collected during a national lockdown, the question remains whether it would be feasible to experimentally induce similar levels of constraints in a laboratory setting. Future studies can focus on boredom experience during more typical daily situations.

## 6. Conclusions

The ability to keep the self entertained appears to provide protection from experiencing the aversive emotion of boredom. Some people reported thriving during the lockdown period, enjoying creative and individual activities that they found fulfilling. Others reported suffering and feelings of isolation and meaninglessness. The question remains about whether people can learn to entertain themselves and if this is a skill that, when developed, provides protection in situations that involve isolation or constraint.

## Figures and Tables

**Figure 1 behavsci-12-00303-f001:**
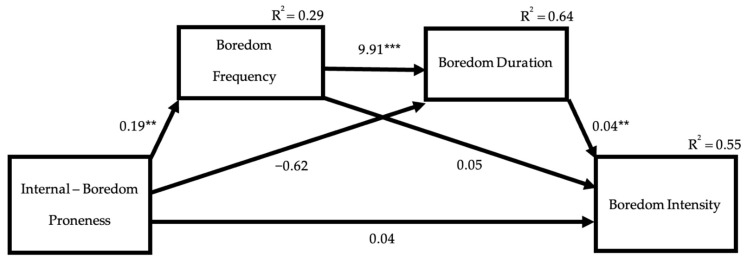
Serial mediation of boredom frequency and boredom duration on the association between the internal factor of boredom proneness and boredom intensity. ** *p* < 0.01, *** *p* < 0.001.

**Table 1 behavsci-12-00303-t001:** Correlations among boredom proneness and boredom experience.

	BP	BP—Internal	BP—External	Boredom Intensity	Boredom Duration
BP					
BP—Internal	0.76 **				
BP—External	0.75 **	0.22			
Boredom intensity	0.26	0.25	0.08		
Boredom duration	0.45 *	0.31	0.37 *	0.72 **	
Boredom frequency	0.56 **	0.44 **	0.33 *	0.61 **	0.79 **

Note. * *p* < 0.05, ** *p* < 0.01.

**Table 2 behavsci-12-00303-t002:** Regression analysis of the association between boredom duration and other boredom experiences.

Predictor	*B*	*SE*	*β*	*t*	*p*
BP—Internal Stimulation	−0.54	0.42	−0.14	−1.30	0.21
BP—External Stimulation	0.40	0.20	0.19	1.98	0.06
Boredom intensity	7.23	1.90	0.44	3.82	<0.001
Boredom frequency	6.05	1.48	0.54	4.09	<0.001

**Table 3 behavsci-12-00303-t003:** Regression analysis of the association between boredom frequency and other boredom experiences.

Predictor	*B*	*SE*	*β*	*t*	*p*
BP—Internal Stimulation	0.10	0.04	0.27	2.41	0.02
BP—External Stimulation	0.001	0.02	0.003	0.03	0.98
Boredom intensity	−0.02	0.25	−0.01	−0.07	0.94
Boredom duration	0.07	0.02	0.73	4.09	<0.001

**Table 4 behavsci-12-00303-t004:** Regression analysis of the association between boredom intensity and other boredom experiences.

Predictor	*B*	*SE*	*β*	*t*	*p*
BP—Internal Stimulation	0.03	0.04	0.13	0.89	0.38
BP—External Stimulation	−0.03	0.02	−0.26	−1.99	0.06
Boredom duration	0.05	0.01	0.81	3.82	<0.001
Boredom frequency	−0.01	0.16	−0.02	−0.07	0.94

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
