# Peer review of "The Struggle to Entertain Yourself: Consequences of the Internal Stimulation Factor of Boredom Proneness during Pandemic Lockdown"

_behavsci, 2022, doi:10.3390/bs12090303_

Round 1

Reviewer 1 Report

This is a well-written paper that deals with an important topic, namely, how individuals’ experiences of boredom were affected during the COVID-19 lockdown. In examining the effects of the COVID-19 lockdown on boredom, the paper explores how one’s responses to the BPS relate to the frequency, intensity, and duration of boredom. It concludes that one of the factors of the BPS (the internal stimulation factor) predicted the frequency of boredom, that frequency of boredom predicted the duration of the experience of boredom, and finally, that duration of the experience of boredom predicted boredom’s intensity.

I think the paper should be published after the authors carry out some revisions:

Initial hypotheses. The authors should describe in more detail their thinking about their initial hypothesis. In particular, what led them to think that the internal factor is the relevant one? In the introduction, the authors write: “Given the restrictions imposed by the COVID-19 lockdown in the United States, where daily activities such as work and social engagement were severely limited, the internal stimulation factor of boredom proneness is potentially relevant to people’s experience of boredom.” I understand that one might expect that boredom would increase during the COVID-19 lockdown, but I’m not sure why one would necessarily think that this has something to do with the internal stimulation factor (i.e., one’s ability (or inability) to maintain interest and engagement). Indeed, isn’t it more intuitive to think that boredom would increase because people can’t do the things that they usually do? Their ability to maintain interest and engagement is the same—they just can’t find appropriate objects of interest and engagement.

Geographical considerations. One might be concerned that the COVID-19 lockdown is not a monolithic phenomenon. In fact, in certain states in the US, there were very few limitations; in other states, individuals experienced much more constraints. Did the authors take that into consideration? Did they collect the locations of their subjects? And did they consider whether location is a factor? If not, perhaps they can address this point and concern in their discussion of limitations.

Short Boredom Proneness Scale (SBPS). Given recent concerns about BPS, it would have been helpful if the authors included SBPS as one of their measures/instruments. In particular, it would be both interesting and useful to know whether the results of this study can be duplicated if a different scale for assessing the presence of trait boredom is used. Perhaps the authors should say something about this in the limitations section of the paper.

Power of the study. I’m concerned that the study/design was underpowered and I would like to invite the authors to address this issue in more detail. When it comes to correlational analysis, an effect size 0.3 with 80% significance (power) and alpha 0.05 requires a minimum of 84 participants. The final sample for the study was 39 participants (about 1/10th of their desired sample size). So, the correlations between, on the one hand, BP-external and, on the other hand, boredom duration and boredom frequency should not be marked as significant (p<0.05). Right? The same, I think, holds for the correlation between BP-internal and boredom frequency (0.44 **). Given that I’m not very well versed in power analysis, I admit that I might have overlooked something here.

Putting aside the collerational relations between the different variables, were there enough participants to allow the authors to confidently report the findings of their regression analyses?

Pre-registration. I would like to know if the authors pre-registered their study and declared their a priori hypotheses before conducting their analyses. If not, then it would be important for the authors to state that in the paper. (I apologize if they already do that and yet I somehow missed it.)

Reviewer 2 Report

The current study focused on the relationships between external versus internal stimulation boredom proneness and characteristics of boredom experience. The authors paid attention to the pattern of relationships from the boredom among emotional experiences, which had certain research significance. However, there are still some major problems in this paper. More specific comments follow:

1. The author’s definition of boredom proneness in the introduction is not clear. Swinkles (1995) has divided boredom into state boredom and trait boredom. State boredom refers to when an individual in a certain situation produces a comparatively short-lived boring emotional experience at a certain moment; trait boredom, is a state of mind, often referred to as a proneness to be bored with stability and individual differences, primarily motivated by individual’s intrinsic motivation (e.g., meaningless life) causes, imperfect cognitive processes, or inefficiency in attention as important factors in producing “idiosyncratic boredom”. Based on this previous literature, I suggest to the author to better define an operational difference between boredom in general in one’s own life (as I think the author measured) and boredom during the usage as a consequence of the usage itself.

2. In the part of Participants, how was the sample size calculated? Did authors perform a power analysis? Please provide more information.

3. The serial numbers of the subheadings in the second part need to be rearranged.

4. In the Materials and methods, is there literature support for measures of boredom experience? If so, please explain in detail.

5. In the part of Discussion, differences to current literature (i.e. original / new results provided by this study). This refers mostly to sections 4.1.

6. In the references section, less than a quarter of the references is from 2019, so the citations need to be updated.

Round 2

Reviewer 2 Report

I don't have any comments.